# Acute Changes on Left Atrial Function during Incremental Exercise in Patients with Heart Failure with Mildly Reduced Ejection Fraction: A Case-Control Study

**DOI:** 10.3390/jpm13081272

**Published:** 2023-08-18

**Authors:** Marco Alfonso Perrone, Ferdinando Iellamo, Valentino D’Antoni, Alessandro Gismondi, Deborah Di Biasio, Sara Vadalà, Giuseppe Marazzi, Valentina Morsella, Maurizio Volterrani, Giuseppe Caminiti

**Affiliations:** 1Department of Clinical Sciences and Translational Medicine, University of Rome Tor Vergata, 00133 Rome, Italy; marco.perrone@uniroma2.it (M.A.P.); iellamo@uniroma2.it (F.I.); 2Cardiology Rehabilitation Unit, IRCCS San Raffaele, 00163 Rome, Italy; valentino.dantoni@sanraffaele.it (V.D.); gismondi.alessandro@outlook.it (A.G.); debbydibiasio20@gmail.com (D.D.B.); vada.sara21@gmail.com (S.V.); giuseppe.marazzi@sanraffaele.it (G.M.); valentina.morsella@sanraffaele.it (V.M.); maurizio.volterrani@sanraffaele.it (M.V.); 3Department of Human Science and Promotion of Quality of Life, San Raffaele Open University, 00163 Rome, Italy

**Keywords:** left atrial function, heart failure, stepwise exercise

## Abstract

Background: the aim of this study was to assess acute changes in left atrial (LA) function during incremental aerobic exercise in patients with heart failure with mildly reduced ejection fraction (HFmrEF) in comparison to healthy subjects (HS). Methods: twenty patients with established HFmrEF were compared with 10 HS, age-matched controls. All subjects performed a stepwise exercise test on a cycle ergometer. Echocardiography was performed at baseline, during submaximal effort, at peak of exercise, and after 5 min of recovery. Results: HS obtained a higher value of METs at peak exercise than HFmrEF (7.4 vs. 5.6; between group *p* = 0.002). Heart rate and systolic blood pressure presented a greater increase in the HS group than in HFmrEF (between groups *p* = 0.006 and 0.003, respectively). In the HFmrEF group, peak atrial longitudinal strain (PALS) and conduit strain were both increased at submaximal exercise (*p* < 0.05 for both versus baseline) and remained constant at peak exercise. Peak atrial contraction strain (PACS) did not show significant changes during the exercise. In the HS group, PALS and PACS increased significantly at submaximal level (*p* < 0.05 for both versus baseline), but PALS returned near baseline values at peak exercise; conduit strain decreased progressively during the exercise in HS. Stroke volume (SV) increased in both groups at submaximal exercise; at peak exercise, SV remained constant in the HFmrEF, while it decreased in controls (between groups *p* = 0.002). Conclusions: patients with HFmrEF show a proper increase in LA reservoir function during incremental aerobic exercise that contributes to maintain SV throughout the physical effort.

## 1. Introduction

In chronic heart failure (HF), the left atrium (LA) undergoes profound structural and electrophysiological changes, ultimately leading to LA dysfunction and dilation [1]. These changes occur in HF patients with both reduced ejection fraction (HrEF) as well as in those with preserved EF (HFpEF) and define the so-called atrial cardiomyopathy [2,3,4,5,6,7]. LA dysfunction is part of the pathophysiological processes that characterize HF, and it contributes to the onset of symptoms in these patients [8,9].

LA function can be assessed through two-dimensional speckle-tracking echocardiography that studies the deformation of atrial walls during the cardiac cycle. This technique allows for distinguishing three atrial phases: reservoir strain, a phase of LA expansion that occurs during left ventricular (LV) systole, conduit, and contraction strain, corresponding, respectively, to the early and late phases of LV diastole [10].

Under physiological conditions, LA function increases during exercise, contributing to the enhancement of LV performance by the Frank–Starling law and ultimately to the increase in cardiac output (CO) [11]. However, the relative LA contribution to CO decreases at a high level of physical efforts when further increases in CO rely mostly on heart rate (HR), which increases linearly with exercise loads [12].

According to the limited currently available literature, in patients with HFrEF and HFpEF, the capacity to enhance LA function in response to increasing external loads is severely impaired [13,14] and the compromission of LA function appears to be a major factor limiting exercise tolerance in these patients [15,16,17].

Since 2016, a third category of HF was introduced by European guidelines and it includes patients with HF with mildly reduced EF (HFmrEF) [18]. This category of HF is largely unexplored compared with HFrEF and HFpEF. In particular, data on LA function in HFmrEF are scant and inconclusive. A reduced LA reservoir strain was described in HFmrEF patients and it was more reduced than in patients with HFpEF [19]. It was also suggested that a reduced LA reservoir strain could be a marker of decreased peak exercise capacity in HFmrEF too [20]. Recently, our group showed that in HFmrEF patients, LA function, despite being reduced at rest compared to normal values, properly increases in response to eccentric isometric resistance exercises. Moreover, there are no studies evaluating the LA response to an incremental stepwise exercise in HFmrEF [21].

The purpose of the present study was to assess the LA response to an incremental symptom-limited aerobic exercise, performed on a cycle ergometer, in patients with post-ischemic HFmrEF in comparison to a control group.

## 2. Materials and Methods

Population. The study enrolled twenty stable patients with HFmrEF secondary to coronary heart disease (CHD) and ten healthy subjects (HS) as control. Patients with HFmrEF were recruited during outpatient visits at the rehabilitation facility of the San Raffaele IRCCS in Rome. All of them previously experienced mild symptoms of heart failure (dyspnea during efforts) but were asymptomatic at the moment of the recruitment and under optimal pharmacological therapy for their disease. These visits were made before starting a cardiac rehabilitation protocol and patients were referred to us by their cardiologists or primary care physicians. During the visits, each patient underwent anthropometric assessment, heart rate (HR) and arterial blood pressure (BP) measurement at rest, echocardiography, and a symptom-limited ergometric test.

The following inclusion criteria were adopted: age between 35 and 75 years; previous diagnosis of CHD; stable sinus rhythm; stable clinical conditions (having no hospitalizations in the last six months; no changes in drug therapy in the last three months); NYHA class I; LV ejection fraction between 40 and 49%; and LA volume index lower than 34 mL/m^2^. The following exclusion criteria were adopted: high blood pressure at rest (after two repeated measurements); uncontrolled arrhythmias at rest; symptomatic heart valve diseases; hypertrophic obstructive cardiomyopathy; symptomatic peripheral arterial disease; neurological and/or orthopedic contraindications to or limiting exercise; and severe chronic obstructive pulmonary disease (FEV1 < 50%). Patients with poor acoustic windows, those with ischemic ECG pattern and/or symptoms suggestive of cardiac ischemia during exercise, those with frequent uncontrolled arrhythmias, and poor exercise performance below 4 metabolic equivalents (METs) were ruled out. Healthy subjects were chosen between workers of the clinic if they fulfilled the following criteria: having an age between 35 and 75 years; no history of cardiac diseases; low cardiovascular risk; no structured physical activities in the previous 6 months; and good acoustic window.

The study complied with the Declaration of Helsinki and was approved by the local Ethics Committee of SanRaffaele IRCCS (protocol number 27/2021). All patients gave written informed consent before entering the study and performed the experimental session within a week of the initial visit. 

Experimental sessions were conducted in the ergometry room of the rehabilitation facility of the San Raffaele IRCCS in Rome. The room temperature was set at 24 °C. All subjects performed an incremental stepwise test on a cycle ergometer (Mortara Instrument, Casalecchio Di Reno, Italy). 

The exercise was initiated at a load of 20 W, and the load was increased every three minutes by 20 W. The pedaling frequency was set at around 60 revolutions per minute during the entire exercise; the exercise was interrupted when volitional exhaustion occurred. During the exercise, HR was continuously monitored and BP was measured at rest at every stage of the incremental exercise and at the first and the fifth minute of recovery. Within the experimental session, echocardiography assessments were performed: (1) at rest; (2) at submaximal effort (50–60% HR max); (3) at peak exercise; and (4) after five minutes of recovery. Rate of pressure product was calculated as the index of myocardial oxygen consumption according to the formula: RPP = (HR × SBP)/100. All experimental sessions were performed in the morning between 10:00 and 12:00 a.m. Subjects were asked to not perform significant physical activities (running, swimming, and cycling) during the 24 h before the experimental session. Subjects were also asked not to smoke or drink wine or other alcoholic beverages in the previous 24 h. They were allowed to have a light meal at least two hours before the start of the experimental session. The research team attending each experimental session was composed of a cardiologist, a nurse, and a cardiac sonographer. 

Echocardiography: All echocardiographic examinations were conducted by an experienced sonographer with subjects in a sitting position. A cardiovascular ultrasound Vivid E95^®^ (GE Healthcare, Chicago, IL, USA) with a 4.0 MHz transducer was used for all examinations. All the echocardiographic images were first digitally stored and then reviewed offline. During the review process, an experienced technician performed deformation measures using a proprietary software (version 10.8, EchoPAC; GE Vingmed Ultrasound, Norway). Left ventricular end-diastolic volume (LVEDV) and end-systolic volume (LVESV) were calculated from the apical two and four-chamber windows using a modified Simpson’s method; stroke volume (SV) was calculated as EDV – ESV, cardiac output (CO) as HR × SV, and ejection fraction (EF) as (EDV − ESV)/EDV. LA volume was measured from a standard apical 4-chamber view at end systole right before mitral valve opening. The biplane method of disks was used to calculate LA volume. LA volume index (LAVI) was calculated by dividing the LA volume by body surface area of the subjects. The reported E/A ratio represents the ratio of peak left ventricle filling velocity in early diastole (E wave) to that in late diastole during atrial contraction (A wave). E/e’ ratio was calculated as the ratio between E wave velocity and the average between lateral and septal wave velocities. Peak systolic LV longitudinal strain and strain rates were assessed using standard apical 4-chamber and 2-chamber views using speckle-tracking analysis. LV global longitudinal strain (GLS) was measured through 2-, 3-, and 4-chamber views. Colour tissue Doppler tracings were obtained with the range gate placed at the lateral mitral annular segments in the apical 4-chamber view. The detection of the LV endocardial boundary was automatically provided by the software; however, the sonographer could edit the measurements to conform to the visualized LV boundaries if necessary. The software generated the longitudinal strain curves for each LV segment and a mean curve for all the segments. The maximum negative value of strain during systole represented the maximum contractility for each segment. The average of these values from each segment then was used to calculate LV GLS. Longitudinal strain measurements were subdivided into a LA reservoir strain, conduit strain, and contractile strain [10]. The reservoir phase was expressed as peak atrial longitudinal strain (PALS), measured at the end of the reservoir phase (positive peak during LV systole); the contraction phase was expressed as peak atrial contraction strain (PACS), measured before the beginning of the active contractile phase (positive peak during early diastole) (Figure 1).

### Statistical Analysis

Data are expressed as mean ± SD. The assumption of normality was checked using the Shapiro–Wilk hypothesis test. Pre- and post-exercise data of normally distributed variables were assessed using repeated measures two-way ANOVA, with Bonferroni corrections for post hoc testing. The two-way ANOVA was used since the study considered two variables as predictors of changes in atrial function: (1) rest or exercise at different intensities; (2) presence or not of HFmrEF. Variables not distributed normally were assessed using the Kruskal–Wallis test and Bonferroni corrections for post hoc testing. The level of significance was set at *p* < 0.05. Data were analyzed using SPSS software (version 20.0 IBM Corp, Amonk, New York, NY, USA).

## 3. Results

Anthropometric and clinical features of patients are summarized in Table 1. HFmrEF patients and HS were matched for age, body mass index, and waist circumference. All patients with HFmrEF were taking beta blockers and ACE-I or ARBs; 85% had a previous myocardial infarction and 56% had a previous coronary artery bypass graft surgery. The reason for stopping the exercise was muscle exhaustion in every patient of this group and no cases of exercise-induced cardiac ischemia occurred. Among HS subjects, two had a diagnosis of pre-hypertension, but they were not taking any medications at the time of the study. At rest PALS, LVGLS, SV, and CO values were significantly lower in HFmrEF compared to HS. HS subjects presented higher resting HR and systolic BP than HFmrEF. 

Submaximal exercise: compared to resting values, PALS increased significantly in both groups without between-group differences (HS = +21.3%; HFmrEF = +23.8%; between groups *p* = 0.354) (Figure 2). PACS presented a greater increase in HS compared to HFmrEF (+50.7% vs. +13.0%, respectively; between groups *p* = 0.001). Conduit strain increased in HFmrEF while it decreased in HS (Figure 3). LVGLS presented small increases in both groups without between-group differences (HS = +6%; HFmrEF = +6.5%; between groups *p* = 0.234). SV increased significantly in both HFmrEF (+10.6%) and HS (+11.4%) groups without between-group differences. LVEDV increased in HFmrEF and HS. HR and CO increased in both groups with a significant greater increase in the HS group than in HFmrEF. E/A ratio and E/e’ ratio did not change compared to baseline values in both groups.

Peak exercise: compared to submaximal values, PALS and conduit strain decreased significantly in HS (−17.5% and −31.2%, respectively) while they remained unchanged in HFmrEF (between groups *p* = 0.025 for PALS and 0.004 for a higher value of METs at peak exercise than HFmrEF (7.4 vs. 5.6; between group *p =* 0.002).

## 4. Discussion

In this study, we assessed acute changes in LA function occurring during an incremental, stepwise, symptom-limited exercise in patients with HFmrEF and in healthy controls. We found that, in HFmrEF, PALS values increased significantly at submaximal exercise and remained constant at peak exercise. Conversely, in HS, PALS values increased at submaximal exercise but then decreased at peak exercise, returning approximately to resting values.

Our results differ from those obtained by other authors in patients with HF [12,14]. In the study of Sugimoto et al. [12] patients with HFpEF presented only small increases in LA reservoir strain during incremental exercise and no increases at all were detected in patients with HFrEF. Tan et al. [14] assessed the acute LA response to exercise in hypertensive subjects and in patients with HFpEF. They observed that the capacity of enhancing atrial function in response to incremental exercise was preserved in hypertensive subjects, while it was lost in patients with HFpEF.

Interestingly, our data comply with the results of a previous study, published by our group and conducted in patients with HFmrEF, in which LA acute response to eccentric resistance training at two different intensities was investigated. In that study, PALS increased at the end of both exercise protocols in comparison to rest values, while no significant exercise-related increases in E/e’ ratio were observed [21].

In the present study, we selectively focused on patients with HFmrEF for two main reasons: firstly because this is a very recently identified subgroup of HF patients in which hemodynamic parameters were less extensively studied in comparison with HFrEF and HFpEF. Secondly, since subjects with HFmrEF enrolled in this study had normal atrial size and were in sinus rhythm, they offer, in our opinion, the opportunity to investigate the atrial remodeling process at its very early stage, in which it consists only of changes in functional parameters that can be non-invasively assessed through speckle-tracking echocardiography. We found that resting values of PALS in patients with HFmrEF were significantly lower compared to HS as well as to reference values [22,23]. The reduced resting values of the reservoir strain could be a sign of an overstretched LA and they were an expected result since it is known that LV chronic ischemia and LV dysfunction both increase LA stiffness and impair LA reservoir function [24].

It should be noted, however, that E/e’ ratio at rest was within the normal range, and similarly to what happened in HS, it did not increase during exercise. Moreover, LAVI was also normal in HFmrEF patients enlisted in this study. Therefore, our results suggest that the impairment of LA reservoir strain was the only marker of LA remodeling detectable in HFmrEF.

Our data comply with other research, underlining that a reduced LA reservoir strain is an early marker of elevated LV filling pressure that precedes the onset of other echocardiographic signs of LV diastolic dysfunction [25,26,27,28]. We observed that during the exercise, PACS increased significantly only in HS. Conversely, in HFmrEF, the trend of PACS during the exercise was represented by a flat curve. The failure to increase PACS in HFmrEF could have different explanations. The inhibition of the sympathetic drive via beta-blockers seems to be a reasonable cause. This hypothesis would be in agreement with a recent study, performed in healthy subjects and suggesting that the contraction strain is modulated mostly by autonomic system activity while it is independent by preload conditions [29]. Alternatively, it could be related to the loss of the elastic properties of the already overstretched atrial myocytes, and together with the reduced PALS values at rest, it could be an early sign of the LV diastolic dysfunction [30]. However, our results do not clarify the mechanisms underlying the absence of an increase in the PACS during exercise in HFmrEF and this aspect of the present research deserves more focused investigations.

In this study, we observed that, at peak exercise, LV diastolic filling decreased in the HS values, while it remained constant in the HFmrEF group when compared to submaximal effort. In the HS group, HR increased more than in HFmrEF both at submaximal and at peak exercise. This is an expected result, since all HFmrEF patients were treated with beta-blockers that curbed their exercise-related increase in HR. The high values reached by HR in HS possibly reduced LA emptying by shortening the diastolic LV filling time. This seems to be confirmed by the decrease in conduit strain values that we observed at peak exercise in such patients. An increased residual blood volume at the atrial level possibly impaired the ability of LA fibers to expand during LV systole to receive further blood from pulmonary veins, causing a reduction in LA reservoir function. The decrease in PALS at peak exercise could in turn explain the parallel reduction in LVEDV, LVGLS, and SV that we registered at peak exercise in comparison to submaximal exercise in HS. This result is in agreement with other research assessing LA and LV dynamics during the exercise of healthy subjects [31,32]. Regarding patients of the HFmrEF group, the constant trend of LV diastolic filling between sub-maximal and maximal effort could explain why LVEDV, SV, and LV GLS increased at submaximal exercise and did not change at peak exercise; moreover, it suggests that LV filling pressure did not change during the different phases of the exercise. This last point seems to be confirmed by the lack of significant increase in E/e’ ratio during the exercise in HFmrEF. Overall, our data let us hypothesize that diastolic LV compliance was at least in part preserved in our HFmrEF patients and that the LA/LV system was working according to the Frank–Starling law during the entire duration of the exercise.

The increase in LA function parameters that we observed in this study can be interpreted as a sign that, despite presenting low resting values of PALS, patients with HFmrEF were still able to mobilize the LA functional reserve during exercise. Clearly, further studies with larger sample sizes are needed in order to confirm our findings. We believe that the results of the present study contribute to the knowledge of the HFmrEF cardiac physiology, and that could be used to develop individualized training protocols for this type of patients.

Limitations: This study has several limitations, the most important being the small sample size that makes it necessary for our results to be verified in larger clinical trials. The present research was conceived as a case-control study between HFmrEF patients and healthy age-matched controls; therefore, patients with HFrEF or HFpEF were excluded. This prevented us from comparing exercise-induced changes in atrial function among different subgroups of HF patients. In our opinion, further studies involving also patients with reduced and preserved EF are needed in order to better understand the findings of the present study The study enrolled mainly male subjects; therefore, our results cannot be generalized to female gender. The high ratio between males and females in the HFmrEF group was caused by the difficulty to find female patients willing to participate to the study. Regarding the control group, we tried to maintain a ratio between the two genders similar to that of the experimental group. In this study, patients were asked to eat a light meal and to avoid alcohol and smoke before the experimental sessions; however, no specific indication regarding the food composition was given. The lack of strict diet control over subjects enrolled in this study is an important limitation since the caloric intake close to the experimental sessions could affect some parameters collected during the exercise. In this research, baseline anthropometric features of patients were limited to BMI, and waist circumferences. However, the body composition was not evaluated. The lack of body composition assessment limits our understanding of characteristics of HFmrEF subjects. In this study, we did not use the propensity score [33] to select the controls, and this possibly increased the risk of selection bias.

All HFmrEF patients had an underlying diagnosis of CHD, and our results may not be valid for patients with non-ischemic HF. Lastly, we enrolled asymptomatic patients at an early stage of LA dysfunction with normal LA size and normal E/e’ ratio; therefore, our results should be applied only to HFmrEF patients with the aforementioned characteristics.

## 5. Conclusions

In this study, patients with HFmrEF presented a proper increase in LA function during an incremental, symptom-limited exercise, without significant rise in LV filling pressure. The enhanced LA performance contributed to the increase SV and CO during the exercise. Further research is needed in order to confirm and extend the results of the present study.

## Figures and Tables

**Figure 1 jpm-13-01272-f001:**
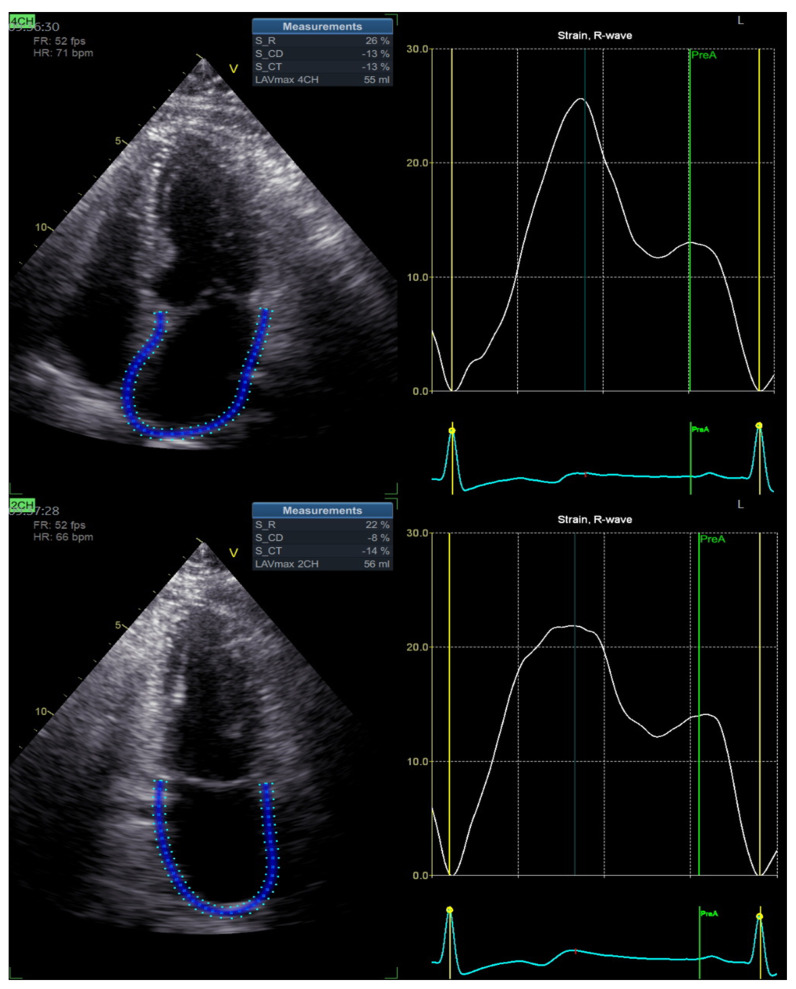
Left atrial strain assessment. LA endocardial borders are automatically tracked by the software in four chamber (4CH) and two chamber (2CH) (blue lines).

**Figure 2 jpm-13-01272-f002:**
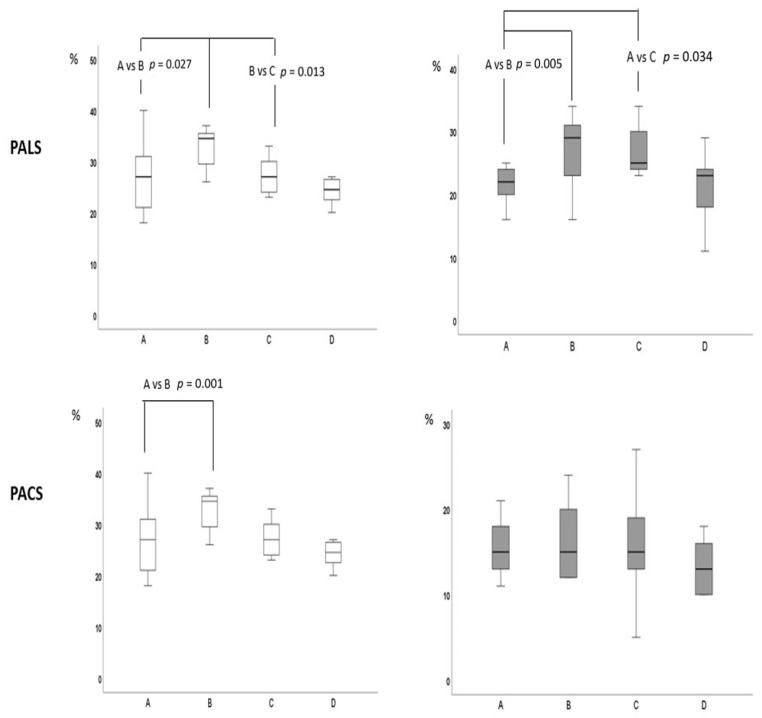
Trends of PALS and PACS during incremental exercise in HS (white boxes) and HFmrEF (gray boxes). Data are reported at baseline (A), submaximal effort (B), peak exercise (C), and recovery (D). Statistical comparisons are made by two-way ANOVA.

**Figure 3 jpm-13-01272-f003:**
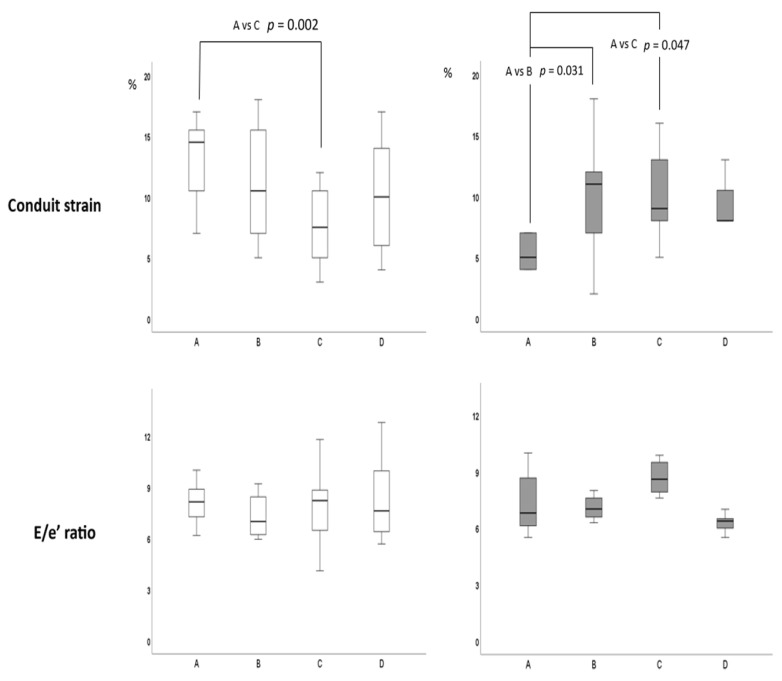
Trends of conduit strain and E/e’ ratio during incremental exercise in HS (white boxes) and HFmrEF (gray boxes). Data are reported at baseline (A), submaximal effort (B), peak exercise (C) and recovery (D). Statistical comparisons are made by two-way ANOVA.

**Table 1 jpm-13-01272-t001:** Anthropometric and echocardiography features of HFmrEF patients and healthy subjects.

	HFmrEF (N = 20)	HS (N = 10)	*p*
Anthropometric characteristics			
Age, years	51.4 ± 10.3	53.3 ± 12.5	0.342
Males/females, *n*	18/2	8/2	0.422
BMI, kg/m^2^	27.8 ± 6.3	28.1 ± 7.4	0.277
Waist circumference, mm	106.8 ± 16.1	108.0 ± 14.3 *	0.351
Previous AMI, *n* (%)	17 (85)	-	
Previous PCI/CABG, *n* (%)	16 (80)/13 (65)	-	
Resting HR, bpm	72 ± 16.3	91 ± 12,4	0.006
Systolic BP, mmHg	107 ± 27.6	129 ± 34.4	0.011
Diastolic BP, mmHg	72 ± 13.6	78 ± 15.2 *	0.344
Comorbidities			
Hypertension, *n* (%)	18 (88)	-	
Diabetes, *n* (%)	4 (37)	-	
Hypercholesterolemia, *n* (%)	19 (82)	3 (30)	0.004
Previous smoke habit/Current smokers, *n* (%)	12 (60)/0 (0)	3 (30)/2 (20)	
Obstruptive sleep apnoea, *n* (%)	2 (10)	-	
Echocardiography parameters			
PALS, %	20	27	0.002
PACS, %	14	15	0.318
LAVI, mL/m^2^	28.6 ± 5.1	27.2 ± 4.9	0.042
E/A ratio	0.97 ± 0.3	0.95 ± 0.6 *	0.128
E/e’ ratio	9.1 ± 2.2	8.0 ± 3.1	0.240
TRV, m/s	2.38 ± 0.3	2.41 ± 0.1	0.131
LV GLS, %	−11.6 ± 4.7	−19.3 ± 3.2	0.028
LVEDV, mL	88 ± 16.8	89.4 ± 14.2	0.086
LVESV, mL	47.2 ± 9.3	39.3 ± 13.5	0.044
LVEF, %	46.1 ± 6.7	56.1 ± 11.4	0.018
SV, mL	41.± 14.8	50.1 ± 12.3	0.003
CO, mL/min	3034.3 ± 189.5	4550.8 ± 232.8	0.001
Treatment			
Antiplatelet therapy, *n* (%)	20 (100)	-	
ACE-Is/ARBs, *n* (%)	20 (100)	-	
Beta blockers, *n* (%)	20 (100)	-	
MRAs, *n* (%)	7 (35)	-	
Nitrates, *n* (%)	3 (15)	-	
Furosemide *n* (%)	4 (20)	-	
Statins, *n* (%)	16 (80)	-	

PCI = percutaneous coronary intervention; CABG = coronary bypass graft; HR = heart rate; PALS = peak atrial longitudinal strain; PACS = peak atrial contraction strain; LAVI = left atrial volume; TRV = tricuspidal regurgitation velocity; LVEDV = left ventricular end diastolic volume; LVESV = left ventricular end systolic volume; LVEF = left ventricular ejection fraction; SV = stroke volume; and CO = cardiac output. * Not-normally distributed variables.

## Data Availability

The data presented in this study are available on request from the corresponding author.

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
