# Peer review of "Acute Changes on Left Atrial Function during Incremental Exercise in Patients with Heart Failure with Mildly Reduced Ejection Fraction: A Case-Control Study"

_jpm, 2023, doi:10.3390/jpm13081272_

Round 1

Reviewer 1 Report

I read with great interest the article entitled « Acute changes on left atrial function during progressive incremental exercise in patients with heart failure with mildly reduced ejection fraction. A case-control study», that concerns an important and clinically interesting issue. I have few comments and suggestions regarding this manuscript.

Minor:

1.      It would be more interesting to compare not only with the control healthy group, but also with HFpEF and/or HFrEF groups. In any case, a more detailed explanation of the choice of patients with HFmrEF is required.

2.      Line 51 – please add a link

3.      It is reasonable to indicate why only patients with NYHA class I participated in the study. There is a high probability that a number of patients did not have heart failure.

4.      In the Materials section, it is advisable to give an echocardiographic imaging with an example of left atrial deformation indices.

5.      What explains such a high resting heart rate (91) in the control group?

6.      The figure and table show only the average values of the LA strain indices. Please indicate the interquartile range/ standard deviation scatter of these indices at each stage of the exercise.

7.      Line 179 – correct the form of heart failure.

8.      What were the causes of exercise cessation in patients in the heart failure group? Was myocardial ischemia detected in this group during exercise? Please insert your answer in the Results section.

9.      It would be interesting to add the dynamics of diastolic stress test indices – TR velocity, E/E' ratio

10.  The Results section does not present the E/A ratio, although it is mentioned in the Materials section.

11.  Please provide an alternative explanation for the decreased reservoir strain at peak exercise only in healthy controls. The assumption of a useful role of chronotropic insufficiency in LV filling does not look convincing.

Minor editing of English language required

Author Response

Reviewer 1

I read with great interest the article entitled « Acute changes on left atrial function during progressive incremental exercise in patients with heart failure with mildly reduced ejection fraction. A case-control study», that concerns an important and clinically interesting issue. I have few comments and suggestions regarding this manuscript.

Minor:

  1. It would be more interesting to compare not only with the control healthy group, but also with HFpEF and/or HFrEF groups. In any case, a more detailed explanation of the choice of patients with HFmrEF is required.

We agree with this comment. We added the lack of patients with HFpEF and HFrEF as a limitation of the present study. Moreover in the discussion paragraph we explained more in detail the choice of patients with HFmrEF

  1. Line 51 – please add a link

Thank you we added a new citation

Agostoni P, Vignati C, Gentile P, et al. Reference Values for Peak Exercise Cardiac Output in Healthy Individuals. Chest.2017;151:1329-1337

  1. It is reasonable to indicate why only patients with NYHA class I participated in the study. There is a high probability that a number of patients did not have heart failure.

Thank you for this comment All subjects recruited in the HFmrEF group had previously experienced mild symptoms of heart failure(dyspnea during efforts)  and were in full HF therapy. However they were asymptomatic at the moment of the recruitment. We specified better this point in the method paragraph

  1. In the Materials section, it is advisable to give an echocardiographic imaging with an example of left atrial deformation indices.

Thank  you, we added the required image

  1. What explains such a high resting heart rate (91) in the control group?

Thank you for this question. There is no a specific reason  for that finding. We hypothesized that it was related to diet factors.  Since there was not a strict control of the diet, we did not verify the real consumption of coffee or others beverages among participants  close to the experimental session. This point has been now specified in the limitations paragraph

  1. The figure and table show only the average values of the LA strain indices. Please indicate the interquartile range/ standard deviation scatter of these indices at each stage of the exercise.

Thank you for this comment. We changed figures accordingly.

  1. Line 179 – correct the form of heart failure.

 Thank you! Done

  1. What were the causes of exercise cessation in patients in the heart failure group? Was myocardial ischemia detected in this group during exercise? Please insert your answer in the Results section.

     Thank you for this comment. In the result section of  the revised version of the paper we specified these points

  1. It would be interesting to add the dynamics of diastolic stress test indices – TR velocity, E/E' ratio

We added data of E/E’ ratio during exercise (in a new figure). Conversely TR velocity was available only in the baseline echocardiography performed at rest (we reported this value in table 1)

  1. The Results section does not present the E/A ratio, although it is mentioned in the Materials section.

Thank you for this comment. We added data of E/A ratio in the result paragraph and in table 1.

  1. Please provide an alternative explanation for the decreased reservoir strain at peak exercise only in healthy controls. The assumption of a useful role of chronotropic insufficiency in LV filling does not look convincing.

Thank you. We accept this comment. in the discussion paragraph we changed our hypothesis

Reviewer 2 Report

The present study is about evaluating the echocardiographic effects during incremental effort in patients with heart failure with slightly reduced ejection fraction. It is a case-control study, with a series of positive points for its execution. One of them that I would like to highlight is the intention to study this relatively new classification of heart failure, as it is of strong interest to cardiologists and professionals in the field of cardiology. However, the study has some elements that affect the final quality of the work presented. I will list below the main points that I observed when reading the study:

1. The authors selected a sample with a lower ratio of women to men in the healthy group. For this type of experimental design, it would be interesting to include the same proportion of participants of different genders.

2. The authors do not describe or evaluate food consumption close to the test of the appraised ones. It was only mentioned that they were instructed to have a “light” meal before the exercise test. Diet can have a significant effect on these variables, especially when compared with cases and controls.

3. The absence of body composition assessment hinders a better understanding of the anthropometric condition of the participants. Only BMI and waist circumference were measured. For patients with heart failure (with or without preserved ejection fraction) this information is insufficient to characterize this new type of heart failure.

4. The representation of the graphs for each variable and for the cases and controls should be improved to value the data. In my opinion, this portion of the text should be revisited and presented in another format that is more assertive to the reader.

Author Response

Reviewer 2

The present study is about evaluating the echocardiographic effects during incremental effort in patients with heart failure with slightly reduced ejection fraction. It is a case-control study, with a series of positive points for its execution. One of them that I would like to highlight is the intention to study this relatively new classification of heart failure, as it is of strong interest to cardiologists and professionals in the field of cardiology. However, the study has some elements that affect the final quality of the work presented. I will list below the main points that I observed when reading the study:

The authors selected a sample with a lower ratio of women to men in the healthy group. For this type of experimental design, it would be interesting to include the same proportion of participants of different genders.

We thanks the reviewer for this comment . Regarding  the control group we tried to maintain a ratio  between male and female genders similar to that of the the experimental group. The high ratio M/ F in the HFmrEF group was because it was difficult  to find female patients willing to participate to the study. This point is better specified in the revised version of the paper

  1. The authors do not describe or evaluate food consumption close to the test of the appraised ones. It was only mentioned that they were instructed to have a “light” meal before the exercise test. Diet can have a significant effect on these variables, especially when compared with cases and controls.

We understand the reviewer concerns  and we admit that the lack of diet control over subjects enrolled in this study is an important limitation since the meals near the experimental sessions could have affected  some parameters collected during the exercise. We included this point among limitations in the  revised version of the paper.

The absence of body composition assessment hinders a better understanding of the anthropometric condition of the participants. Only BMI and waist circumference were measured. For patients with heart failure (with or without preserved ejection fraction) this information is insufficient to characterize this new type of heart failure.

We agree with this point underlined by the reviewer  and we admit  that the assessment  of body composition could have  further characterized our sample. Unfortunately these evaluation have not been performed. We included this point among limitations in the  revised version of the paper.

  1. The representation of the graphs for each variable and for the cases and controls should be improved to value the data. In my opinion, this portion of the text should be revisited and presented in another format that is more assertive to the reader.

Thank you . In order to improve data presentation we modified figure 1 and .e added a new figure with  trends of conduit strain and E/e’ ratio.

Reviewer 3 Report

The article has made significant progress in corrections with suggestions and there are a few points that need to be mentioned:

- The relatively small number of patients should be mentioned in the limitations section.

- You can benefit from this titled article "The Benefits of Sacubitril-Valsartan in Low Ejection Fraction Heart Failure. Abant Medical Journal 11 (3), 337-336" in the discussion section.

Author Response

Reviewer 3

The article has made significant progress in corrections with suggestions and there are a few points that need to be mentioned:

- The relatively small number of patients should be mentioned in the limitations section.

Done

- You can benefit from this titled article "The Benefits of Sacubitril-Valsartan in Low Ejection Fraction Heart Failure. Abant Medical Journal 11 (3), 337-336" in the discussion section.

Thank you for this  remarkable paper, however after reading it we could not find a direct relation with data of our manuscript

Reviewer 4 Report

Thanks for asking me to review this study. From a statistical point of view I have the following suggestions:

- Was propensity score used to select the controls.

- Which of the variables in Table 1 were normally distributed and which weren't. Maybe you can indicate the same with symbols and indicate them in the legend.

- For two way ANOVA apart from case/control which was the other variable used as a predictor?

Page 1 : Line 42: Should be "that characterize"

Line 66: Should be "reduced at rest"

Line 186: patients

Is the word betablockers or beta blockers? 

Author Response

Thanks for asking me to review this study. From a statistical point of view I have the following suggestions:

- Was propensity score used to select the controls.

We did not use the propensity score to select controls. In the revised version of the paper we added this point as a limitation of the present study

- Which of the variables in Table 1 were normally distributed and which weren't. Maybe you can indicate the same with symbols and indicate them in the legend.

Thank you, we agree with the reviewer comment ad made these changes in table 1.

- For two way ANOVA apart from case/control which was the other variable used as a predictor?

Thank you for this question. The two variables considered in the study as predictors of changes of atrial function were 1) rest/ different levels of  exercise; 2) presence or not of the disease (heart failure). In the revised version of the paper we specified this point in the statistical paragraph

Page 1 : Line 42: Should be "that characterize"

Thank you. Done

Line 66: Should be "reduced at rest"

Done

Line 186: patients

Thank you. Done

Is the word betablockers or beta blockers? 

Thank you. It was beta blockers. We made the correction 

Round 2

Reviewer 1 Report

-

Minor editing of English language required

Author Response

Thank you for your effort

Reviewer 2 Report

All of my concerns and comments have been properly addressed.

Author Response

Thank you for your effort

Reviewer 3 Report

Corrections are not enough.

Author Response

Thank you for your effort